# Pathological Role of HDAC8: Cancer and Beyond

**DOI:** 10.3390/cells11193161

**Published:** 2022-10-09

**Authors:** Ji Yoon Kim, Hayoung Cho, Jung Yoo, Go Woon Kim, Yu Hyun Jeon, Sang Wu Lee, So Hee Kwon

**Affiliations:** College of Pharmacy, Yonsei Institute of Pharmaceutical Sciences, Yonsei University, Incheon 21983, Korea

**Keywords:** HDAC8, HDAC8 selective inhibitors, epigenetics, cancer

## Abstract

Histone deacetylase 8 (HDAC8) is a class I HDAC that catalyzes the deacetylation of histone and non-histone proteins. As one of the best-characterized isoforms, numerous studies have identified interacting partners of HDAC8 pertaining to diverse molecular mechanisms. Consequently, deregulation and overexpression of HDAC8 give rise to diseases. HDAC8 is especially involved in various aspects of cancer progression, such as cancer cell proliferation, metastasis, immune evasion, and drug resistance. HDAC8 is also associated with the development of non-cancer diseases such as Cornelia de Lange Syndrome (CdLS), infectious diseases, cardiovascular diseases, pulmonary diseases, and myopathy. Therefore, HDAC8 is an attractive therapeutic target and various HDAC8 selective inhibitors (HDAC8is) have been developed. Here, we address the pathological function of HDAC8 in cancer and other diseases, as well as illustrate several HDAC8is that have shown anti-cancer effects.

## 1. Introduction

Epigenetics is the study of changes in gene expression without changes in DNA sequences [1]. The dynamic epigenetic status of chromatin regulates cell development, differentiation, growth, cell cycle, and aging, which collectively affect human health and diseases. Given that the accumulation of epigenetic alterations is heritable and drives the malignant transformation of cells, a better understanding of epigenetic mechanisms may expand potential treatment options for various diseases. Epigenetic treatments are particularly promising because epigenetic alterations are reversible by chemical drugs or gene therapy, reverting the pathological phenotypes to normal [2].

Post-translational modifications (PTMs) of histones include acetylation, phosphorylation, methylation, ubiquitination, and SUMOylation. Among these modifications, histone acetylation notably modifies the chromosomal structure and regulates gene expression [3,4]. Histone deacetylase (HDAC) removes the acetyl group from lysine residues of histones. This results in the formation of condensed heterochromatin and principally suppresses gene expression [5]. HDACs also target non-histone proteins and regulate critical biochemical functions such as protein stability, mRNA stability, and enzyme activity. Regular functions of HDACs are required for hematopoiesis and the development of the heart, muscle, and skeleton. Deregulation of HDACs may cause neurodegenerative diseases, metabolic disorders, and inflammation [6,7,8]. HDAC overexpression is commonly detected in advanced stages of cancer and drives poor outcomes in patients, suggesting that HDACs are involved in multiple stages of cancer [9].

Mammalian HDACs consist of 18 isoforms that are divided into four classes based on their sequence similarity to yeast factors and cofactor dependency [7]. Class I (HDAC1, HDAC2, HDAC3, HDAC8), class IIa (HDAC4, HDAC5, HDAC7, HDAC9), class IIb (HDAC6, HDAC10), and class IV (HDAC11) belong to the classical family that requires Zn^2+^ for the enzyme activity. Class III (SIRT1-7) HDACs are classified as silent information regulator 2 (Sir2)-related proteins (sirtuin) family and require NAD^+^ as a cofactor [10].

Since the identification of HDAC8 as the latest member of class I HDAC in 2000, it has been both structurally and functionally well-studied through advanced cellular experiments [11,12,13]. HDAC8 performs critical roles in human gene transcription and pathophysiology by targeting core histones as well as non-histone substrates, such as p53 and structural maintenance of chromosomes 3 (SMC3) [14,15,16]. This review will explore various functions of HDAC8 and its importance in diseases. In particular, we summarize recent studies of HDAC8 in cancer development and suggest HDAC8 as a potential therapeutic target in human cancers.

## 2. Subcellular Localization and Structure of HDAC8

HDAC8 is a sex-linked gene that is located in the chromosomal position Xq13.1 [13]. The complete sequencing of 377 amino acid residues and phylogenetic analysis of HDAC8 categorized the enzyme to class I HDAC. HDAC8 has over 30% amino acid identity with other class I HDACs, including the nine conserved blocks that serve the catalytic function of deacetylase, but have a shorter C-terminal sequence [11,13]. Originally, HDAC8 was characterized as a nuclear protein [12]. However, various studies have identified the nuclear and cytoplasmic distribution of HDAC8 [17,18,19] and predominant cytoplasmic localization in smooth muscle, brain, and melanoma cells [20,21,22,23]. Li et al. demonstrated that AMP-activated protein kinase (AMPK) phosphorylates HDAC8 and mediates its translocation to the cytoplasm from the nucleus, suggesting that phosphorylation may be used to direct subcellular localization of HDAC8 depending on its function and cellular condition [24].

HDAC8 is the first human HDAC to have its three-dimensional structure experimentally determined by X-ray crystallography (Figure 1) [15]. Structurally, HDAC8 is a head-to-head dimer of two essentially identical molecules. Each molecule possesses one zinc-binding site that promotes catalytic activity and two potassium-binding sites that enhance structural stability. The metal binding sites are interconnected and determine the enzyme activity of HDAC8 [25]. The active site of HDAC8 comprises a hydrophobic tunnel that leads to the catalytic machinery [15]. The tunnel walls consist of six residues (F152, F208, Y306, G151, H180, and M274) that occupy four methylene groups of the substrate. At the end of the tunnel lies the catalytic machinery where the zinc ion bound to D178, D267, and H180 coordinates the deacetylation process. Here, the zinc ion and Y306 activate the amide carbonyl (C = O) of the substrate for a nucleophilic attack. D101 plays a critical role in substrate binding by directing conformational changes of the L2 loop from an unliganded form to a liganded form [26]. 

## 3. Regulation of HDAC8 Expression and Activity

HDAC8 expression can be modulated by regulating the level of transcription, translation, or degradation (Figure 2). In terms of transcription, several transcription factors have been identified to bind to the promoter of HDAC8 and activate transcription. These transcription factors include forkhead box protein O1/3 (FOXO1/3), SRY-box transcription factor 4 (SOX4), cAMP responsive element binding protein 1 (CREB1), sterol regulatory element-binding protein 1 (SREBP-1), and Yin Yang 1 (YY1) [28,29,30,31,32]. HDAC8 translation can be repressed by microRNAs (miRNAs) interacting with the 3′ untranslated regions (3′UTR) of the target mRNA [33]. Specifically, miR-451(a), miR-216b-5p, miR-144-3p, miR-483-3p, miR-21-3p, miR-93, miR-95-5p, and miR-455-3p target HDAC8 mRNA to decrease its expression [34,35,36,37,38,39,40,41,42,43]. Furthermore, HDAC8 expression levels depend on the degradation system. In lung cancer cells, cAMP signaling increases HDAC8 protein levels by inhibiting PI3K-AKT-JNK-mediated autophagy and proteasomal degradation [44]. 

As previously mentioned, AMPK-mediated phosphorylation of HDAC8 regulates its subcellular localization. However, phosphorylation of HDAC8 can also negatively regulate its deacetylase activity. cAMP-dependent protein kinase A (PKA)-mediated phosphorylation of HDAC8 at Ser39 suppresses HDAC8 catalytic activity, both in vitro and in vivo [45]. Nakayama et al. demonstrated that PKA inhibitor H89 suppresses HDAC8 phosphorylation, activating HDAC8 and reducing histone acetylation, whereas PKA activator forskolin hyperphosphorylates HDAC8, suppressing HDAC8 activity and increasing histone acetylation {Nakayama, 2017 #571}.

## 4. HDAC8 Substrates

HDAC8 deacetylates histones preferentially at histone H3 lysine 9 (H3K9) and H3 lysine 27 (H3K27) and regulates the transcription of its target genes. Multiple non-histone substrates such as SMC3, p53, and estrogen-related receptor alpha (ERRα) have also been identified (Figure 3). HDAC8 deacetylates its substrates and regulates protein stability and activity. Although multiple HDAC8 substrates have been identified so far, the development of biochemical investigation techniques potentiates the discovery of other HDAC8 substrates [46,47].

### 4.1. Histone Substrates

HDAC8 deacetylates all core histones in vitro [11,12]. However, following studies indicated limited histone deacetylating activity of HDAC8 in vivo. No significant changes in histone acetylation levels were observed when HDAC8 was overexpressed or inhibited [13,48,49]. It has been suggested that the failure in detecting in vivo histone substrates arises from limitations in experimental methods or from the compensation of other HDACs that mask the deacetylase activity of HDAC8. [14]. Indeed, specific antibodies could detect the distinct sites of histones deacetylated by endogenous and ectopically expressed HDAC8, preferentially at H3K9ac and H3K27ac [50,51,52,53,54]. HDAC8-mediated deacetylation of H3/H4 histones at promoter or enhancer regions is involved in transcriptional repression of several genes such as interferon beta 1 (*IFNB1*)*,* suppressor of cytokine signaling 1/3 (*SOCS1/3*)*,* sirtuin 7 (*SIRT7*)*,* mitogen-activated protein kinase kinase 3 (*MAP2K3*)*,* C-C motif chemokine 4 (*CCL4*)*,* inhibitor of DNA binding 2 (*ID2*), BCL2 interacting protein 3 (*BNIP3*), and metastatic lymph node 64 (*MLN64*) [54,55,56,57,58,59,60] (Table 1). 

### 4.2. Non-Histone Substrates

#### 4.2.1. SMC3

HDAC8 interacts with and deacetylates various non-histone proteins. SMC3 is a unique substrate of HDAC8, first reported in 2012 [61]. SMC3 is a subunit of cohesin that is acetylated at Lys105 and Lys106 by the establishment of sister chromatid cohesion N-acetyltransferase 1/2 (ESCO1/2) and deacetylated by HDAC8 (Table 2) [62]. Deacetylation of SMC3 is crucial for cohesin-related functions such as sister chromatid cohesion, genome organization, and transcriptional regulation. Deacetylation of SMC3 enables proper dissolution of the cohesin complex so that it can be recycled for the following cell cycle. Loss of HDAC8 activity results in the accumulation of acetylated SMC3 and the retention of pro-cohesive elements, consequently disrupting the cell cycle [61]. HDAC8 also plays an important role in chromatin organization by regulating the size of chromatin architectural stripes and chromatin loops. Deacetylation of SMC3 by HDAC8 restarts the looping reaction that was restricted when the chromatin loops were bound to PDS5 cohesin-associated factor A (PDS5A) under SMC3 acetylation [63].

#### 4.2.2. Tumor Suppressors

p53 is often referred to as the “guardian of the genome” and regulates multiple cellular events such as cell cycle progression and apoptosis [64]. Acetylation of p53 is crucial for its stabilization and activation and is mediated by CREB-binding protein (CBP)/p300, CBP/p300-associated factor (PCAF), males-absent on the first (MOF), monocytic leukemic zinc finger (MOZ), and Tat-interactive protein 60 (TIP60). In addition to HDAC8, other HDACs such as HDAC1, HDAC6, SIRT1, and SIRT3 have been previously identified to deacetylate p53 [65]. Due to the pivotal role of p53 as a tumor suppressor, p53 activation through HDAC8 inhibition is a potential therapeutic strategy and has been shown to be effective in ovarian cancer, hepatocellular carcinoma (HCC), and acute myeloid leukemia (AML) [28,66,67,68]. So far, HDAC8 has been identified to deacetylate p53 at Lys381 and Lys382 [66,67,68]. Using combined analysis of chemical tools and proteomics, another tumor suppressor, AT-rich interactive domain-containing protein 1A (ARID1A), has been introduced as an unbiased candidate substrate of HDAC8. HDAC8 inhibition through PCI-34051 treatment greatly reduces ARD1A acetylation and in vitro enzymatic deacetylation experiment showed that recombinant HDAC8 deacetylates synthetically acetylated ARID1A peptide [48].

#### 4.2.3. Kinases

Pyruvate kinase isoenzyme type M2 (PKM2) is a central regulator of signaling pathways that plays a dominant role in cancer development and progression. PTMs such as phosphorylation, oxidation, and acetylation are known to affect PKM2 activity [69]. Accordingly, HDAC8-mediated deacetylation of PKM2 at Lys62 residue regulates its stability, activity, and localization to the nucleus [70]. Furthermore, protein kinase B (PKB), also known as AKT, is a key player in the signal transduction pathways, that is deacetylated by HDAC8 at Lys426. AKT Lys426 to Arg mutant (K426R, lysine deacetylation mimic) increases GSK-3β phosphorylation while AKT Ly426 to Gln mutant (K426Q, lysine acetylation mimic) reduces GSK-3β phosphorylation, indicating that HDAC8-mediated deacetylation is crucial for the phosphorylation and activation of AKT [71].

#### 4.2.4. Cytoskeletal Proteins

Major targets of HDAC6, such as α-tubulin and cortactin, are also substrates of HDAC8 [72,73]. Acetylation of α-tubulin at Lys40 is mediated by alpha-tubulin N-acetyltransferase (αTAT1)/MEC-17 [74,75]. Deacetylation of α-tubulin is mainly mediated by HDAC6, but HDAC8 has also recently been characterized to deacetylate α-tubulin [76]. In HeLa cells, HDAC8 is significantly overexpressed and potentially takes over the function of HDAC6 to become the major deacetylase of α-tubulin [76]. Cortactin is an actin-binding protein that is acetylated by PCAF on nine lysines (Lys87, Lys124, Lys161, Lys189, Lys198, Lys235, Lys272, Lys309, and Lys319) within the repeat region. Acetylation of cortactin has shown to inhibit actin polymerization by decreasing its association with F-actin [73]. HDAC8 was identified to mediate the deacetylation of cortactin, enhancing actin filament polymerization and smooth muscle contraction [77]. HDAC8 is also involved in the regulation of an actin-severing protein, cofilin. HDAC8 deacetylates heat shock protein 20 (HSP20) at Lys160 and interacts with cofilin to form the HDAC8/HSP20/cofilin complex. Although the precise mechanism is yet to be uncovered, Karolczak-Bayatti et al. suggested that acetylated HSP20 reduces inactive phospho-cofilin levels, thereby affecting actin dynamics [21].

#### 4.2.5. Transcription Factors

ERRα is an orphan nuclear receptor (NR) that is mainly involved in cellular metabolism by regulating the transcription of metabolic genes [78]. Acetylation of ERRα is catalyzed by PCAF at four lysines (Lys129, Lys138, Lys160, and Lys162) located within the zinc finger and C-terminal extension (CTE) of the DNA-binding domain (DBD). ERRα acetylation reduces its DNA-binding and transcriptional activity. HDAC8 directly deacetylates ERRα, thereby reactivating its transcriptional activity [79]. c-Jun is an oncogenic transcription factor that belongs to the activating protein 1 (AP-1) family [80]. Among three potential acetylation sites (Lys268, Lys271, and Lys273), HDAC8 deacetylates c-Jun at Lys273 and increases its transcriptional activity in melanoma cells [81]. HDAC8 also interacts with other binding partners, such as CREB and RUNX family transcription factor 2 (RUNX2) and represses their activities [50,82].

**Table 2 cells-11-03161-t002:** HDAC8 substrates and interacting partners.

	Protein	Deacetylation Site	Effect	Related Disease	Reference
**Substrate**	SMC3	Lys105 and Lys106	Recycling of cohesin complex Chromatin organization	CdLS	[61,63,83]
	p53	Lys381 and Lys382	Repressed stability and activity	Ovarian cancer HCC AML	[28,66,67,68]
	ARD1A	N/A	N/A	N/A	[48]
	PKM2	Lys62	Nuclear localization Glucose metabolism	HCC	[70]
	AKT	Lys426	Enhanced stability and activity	BC	[71]
	α-Tubulin	Lys40	Deregulation of microtubule structural organization Cell migration	Cervical cancer Glioma DMD	[76,84,85]
	Cortactin ^1^	Nine lysines within the repeat region ^1^	Actin filament polymerization Smooth muscle contraction	N/A	[77]
	HSP20	Lys160	Actin filament polymerization Smooth muscle contraction	Premature birth	[21]
	ERRα ^1^	Four lysines within the DBD ^1^	Enhanced transcriptional activity	N/A	[79].
	c-Jun	Lys273	Enhanced transcriptional activity	Melanoma	[81]
**Interacting Partner**	CREB [82], PP1 [82], RUNX2 [50], Gal-3 [86], CM [68], Cofilin [21], HSP70 [87], EST1B [87], α-SMA [88]				

^1^ Potential HDAC8 deacetylation sites. Further verification is required.

## 5. HDAC8 in Cancer

HDAC8 plays a multifunctional role in cancer progression (Figure 4). By acting on histones and non-histone substrates, HDAC8 stimulates tumor growth and metastasis by enhancing cell proliferation, suppressing apoptosis, and activating epithelial-mesenchymal transition (EMT). HDAC8 is also deeply involved in the process of immune evasion and drug resistance, which are crucial for tumor progression. Accordingly, genetic ablation or pharmacological inhibition of HDAC8 demonstrates anti-cancer effects in various types of cancer. 

### 5.1. Cancer Cell Proliferation and Apoptosis

HDAC8 is overexpressed in cancers such as gastric cancer, HCC, oral squamous cell carcinoma, and childhood acute lymphoblastic leukemia. [67,89,90,91]. Especially, HDAC8 expression is positively correlated with advanced-stage diseases and poor outcomes in neuroblastoma and breast cancer (BC) [71,92]. Conversely, decreased expression of HDAC8 is associated with poor prognosis in metastatic melanoma and intrahepatic cholangiocarcinoma [23,93]. Nevertheless, numerous studies support the tumorigenic role of HDAC8 and confirmed that genetic ablation or pharmacological inhibition of HDAC8 reduces cell proliferation, suppresses colony formation, and induces cell cycle arrest in cancer [25,29,36,84,89,90,92,94,95]. Upregulation of apoptosis is also frequently observed when cancer cells are treated with HDAC8is [94,96,97,98,99]. HDAC8 depletion or inhibition induces p53/p21-mediated apoptosis in HCC cells, Bcl2 modifying factor-mediated apoptosis in gastric cancer and colon cancer cells, and caspase-dependent apoptosis in T cell lymphomas [31,49,89,100]. 

Several studies have investigated the molecular mechanism of HDAC8-induced tumor cell proliferation. Zhang et al. presented that HDAC8 deacetylates the Lys62 residue of PKM2 and facilitates its transport to the nucleus, where it binds to β-catenin, promoting cyclin D1 (*CCN**D1*) gene transcription and cell cycle progression [70]. In myeloproliferative neoplasm (MPN) cells, HDAC8 suppresses the expression of SOCS1 and SOC3. This hyperactivates Janus kinase 2/signal transducers and activators of transcription (JAK2/STAT) signaling and promotes the oncogenesis of MPNs. Consistently, HDAC8 knockdown increases the expression of SOCS1/3 and suppresses cell proliferation [56]. HDAC8 binds to the human ortholog of the yeast ever-shorter telomeres 1B (hEST1B) and protects it from ubiquitin-mediated degradation. Phosphorylated HDAC8 recruits HSP70 to the hEST1B-carboxy-terminus of the Hsc70-interacting protein (CHIP) complex, which inhibits CHIP E3 ligase-mediated degradation of hEST1B. Because telomerase activity is responsible for senescence escape in tumor cells, these results suggest that HDAC8-mediated stability of hEST1B may contribute to increased tumor cell proliferation [87]. 

p53 has been intensively studied in HDAC8-related tumor growth. In AML, the inversion of chromosome 16, inv(16), gives rise to aberrant production of fusion protein CBFβ-SMMHC (CM). CM interacts with both HDAC8 and p53 and mediates the deacetylation and inactivation of p53, promoting leukemia stem cell (LSC) transformation and cell survival. Accordingly, inhibition of HDAC8 induces apoptosis by restoring p53 acetylation and activation [68]. The effect of HDAC8 on p53 is not confined to wild-type p53. Mutant p53 undermines the tumor suppressive function of wild-type p53 and promotes tumor cell survival. In triple-negative breast cancer (TNBC) cells, inhibition of HDAC8 increases the acetylation of YY1, decreasing its transcriptional activity and downregulating mutant p53 transcription, consequently repressing tumor cell proliferation [101]. In another research, HDAC8 knockdown decreased the transcription of mutant p53 by decreasing HOXA5 expression and activity, subsequently inhibiting cell proliferation [102]. Contrary to previous studies, however, Singh et al. reported that HDAC8 activator TM-2-51 enhances the expression of p53, inducing apoptosis and inhibiting cell growth in p53 wild-type neuroblastoma cells. [103]. 

### 5.2. Metastasis 

#### 5.2.1. Breast Cancer Metastasis

HDAC8 is highly expressed in BC relative to other cancers [17]. TCGA database analysis indicated that HDAC8 expression increases with the grade and stage of BC patients. Additionally, BC cell models with overexpressed HDAC8 are more likely to metastasize to the lungs [71]. MDA-MB-231, one of the most invasive BC cell lines, expresses higher levels of HDAC8 than MCF-7, a less invasive BC cell line [104].

HDAC8 triggers BC metastasis in vitro and in vivo. BC migration is enhanced when HDAC8 is overexpressed and significantly suppressed upon HDAC8 knockdown or PCI-34051 treatment [105,106]. HDAC8 downregulation also decreases mesenchymal cell makers such as fibronectin, vimentin, matrix metalloproteinase-9 (MMP-9), and MMP-2. In addition, detyrosination of α-tubulin, an important marker of migration and invasion of cancer cells, is decreased upon HDAC8 inhibition in BC cells. Mice inoculated with MDA-MB-231 stably expressing HDAC8 present a greater number of lung tumors [71]. Furthermore, in 4T1 BC xenograft, PCI-34051 treatment significantly decreases lung metastatic nodules compared to that of the control mouse [57]. 

HDAC8 promotes metastasis through several mechanisms. In TNBC cells, analysis of several migration-related signal factors revealed that yes-associated protein 1 (YAP), an important Hippo pathway target that drives metastasis, is upregulated upon HDAC8 overexpression. HDAC8 stabilizes the expression and localization of YAP by suppressing its phosphorylation at Ser127. Depletion or inhibition of YAP using siRNA or verteporfin, respectively, abolishes HDAC8-induced cell migration, suggesting that HDAC8 regulates TNBC migration by enhancing YAP expression. Interestingly, HDAC8-dependent regulation of YAP was not observed in non-TNBC cells [105]. Investigation of EMT transcription factors, including TWIST, ZEB1, and SNAIL, revealed that only SNAIL is upregulated upon HDAC8 overexpression. PCI-34051 treatment or HDAC8 knockdown decreases the expression of SNAIL and its nuclear localization, while overexpression of SNAIL attenuates EMT suppression effects of PCI-34051. These results indicate that HDAC8 promotes metastasis via the regulation of SNAIL. Further investigation demonstrated that HDAC8 deacetylates AKT, increasing its phosphorylation and activation. AKT, in turn, phosphorylates and suppresses the catalytic activity of GSK-3β. Downregulation of GSK-3β activity represses SNAIL phosphorylation, enhancing protein stabilization and nuclear localization of SNAIL [71]. Transforming growth factor-β (TGF-β) signaling also promotes metastasis in BC. HDAC8 is a co-factor of a positive feedback loop that hyperactivates TGF-β signaling by suppressing SIRT7 transcription. HDAC8 inhibition suppresses BC cell migration induced by TGF-β1 while SIRT7 knockdown attenuates this effect in vivo and in vitro, implicating that HDAC8 inhibition suppresses metastasis by enhancing SIRT7 transcription [57]. These findings show that HDAC8 plays an important role in breast cancer metastasis.

#### 5.2.2. Metastasis in Other Cancers

HDAC8 has been reported to be involved in metastasis of other cancers besides BC. In neuroblastoma, HDAC8 is the only enzyme among 11 HDAC isoforms (HDAC1-11) that significantly correlates with the advanced stage [92]. In cervical cancer cells, PCI-34051 treatment significantly reduces cell migration [76]. In prostate cancer, HDAC8 promotes cancer metastasis by repressing the expression of maspin, a tumor suppressor that regulates cell migration and invasion. Knockdown of HDAC8 rescues maspin expression by increasing p53 acetylation and its binding with maspin promoter, enhancing maspin transcription and significantly reducing migration in prostate cancer cells [107]. HDAC8 promotes glioma migration by regulating the acetylation levels of α-tubulin, which is important in microtubule structural organization and cell migration. HDAC8 inhibition by PCI-34051 treatment elevates acetylated α-tubulin and reduces cell migration in glioma cells [84]. HDAC8 upregulates EMT in HCC by binding to the enhancer region of inhibitor of DNA binding 2 (ID2) and decreasing its transcription. ID2 suppresses EMT through direct interaction with TWIST. Consistently, downregulation of ID2 using ID2-AS1, a long non-coding RNA (lncRNA) that enhances ID2 transcription via blocking HDAC8 occupancy at the ID2 enhancer region, inhibits HCC invasion and migration in vitro and in vivo [59].

### 5.3. Immune Evasion

HDAC8 has been recognized to be involved in several strategies of immune evasion. Glioma creates a tumor microenvironment (TME) that supports tumor growth by switching glioma-associated microglia/macrophages (GAMs) to the anti-inflammatory phenotype as well as impairing natural killer (NK) cell recruitment and activity. HDAC8 inhibition by PCI-34051 treatment switches GAMs to the pro-inflammatory/anti-tumoral phenotype. PCI-34051 treatment also enhances the expression of natural killer group 2D (NKG2D) ligands that trigger the cytotoxic activity of NK cells and increases infiltration of CD69^+^ and NKG2D^+^ NK cells in glioma tumor mass [84]. In mantle cell lymphoma (MCL), HDAC8 inhibition has been observed to increase interferon-gamma (IFNγ)-producing NK cells [99]. 

Immunosuppressive TME and insufficient tumor-infiltrating lymphocytes limit the effectiveness of immune-checkpoint blockade (ICB) in HCC patients. Yang et al. proposed an HDAC8-regulated enhancer program that promotes T cell exclusion from the tumor, attenuating responsiveness to programmed death-ligand 1 (PD-L1) blockade in HCC. Chromatin profiling revealed that HDAC8 silences enhancers of the genes involved in immune-related functions such as T cell–trafficking chemokines via H3K27 deacetylation. Downregulation of HDAC8 restores H3K27 acetylation, promoting T-cell trafficking chemokine production and intratumoral CD8^+^ T cell infiltration. HDAC8 inhibition in combination with ICB therapy using anti–PD-L1 antibodies induces effective and durable anti-tumor responses [54]. Moreover, HDAC8 suppresses anti-tumor immunity in melanoma cells by reducing the expression of PD-L1. HDAC8 inhibition upregulates HOXA5/STAT3-mediated transcriptional activation of PD-L1 encoding gene, *CD274*, thereby enhancing anti-tumor T-cell response in melanoma cells [108]. 

### 5.4. Drug Resistance

#### 5.4.1. Chemotherapy

Paclitaxel resistance in BC patients is largely attributed to TGF-β activation. HDAC8 promotes paclitaxel resistance by hyperactivating TGF-β signaling via SIRT7-SMAD4 axis. HDAC8 inhibition sensitizes BC cells and xenograft mouse models to paclitaxel by rescuing SIRT7 expression. Clinically, patients with high HDAC8 expression show worse prognosis after receiving chemotherapy, further highlighting the importance of HDAC8 in drug resistance [57].

HDAC8 is also associated with temozolomide (TMZ) resistance in GBM cells. HDAC8 is significantly upregulated in TMZ-resistant GBM cell lines compared to the parental GBM cells. NBM-BMX (BMX), an HDAC8i, greatly enhances the cytotoxic effect of TMZ in resistant cells. Combination treatment of BMX and TMZ suppresses cell proliferation by the Wnt/β-catenin/GSK-3β pathway, induces cell cycle arrest in G0/G1, and promotes p53-mediated apoptosis. Furthermore, BMX and TMZ treatment attenuates GBM stem cell characteristics and reduces protein levels of stem cell makers such as CD44, SOX2, and CD133 to overcome TMZ resistance [109].

In neuroblastoma, HDAC8 inhibition increases sensitivity to doxorubicin (Dox). HDAC8 inhibition using siRNA upregulates miR-137 which increases Dox sensitivity in neuroblastoma cells by reducing the expression of constitutive androstane receptor and its target, multidrug resistance protein 1 (MDR1). Subsequently, depletion of HDAC8 downregulates the expression of MDR1 and elevates Dox sensitivity in neuroblastoma cells, while knockdown of miR-137 reverses these effects [110]. 

In AML, HDAC8 promotes daunorubicin (DNR) resistance via regulating interleukin-6 (IL-6) and IL-8 expression. DNR-resistant AML cells show significant overexpression of HDAC8, IL-6, and IL-8. Targeted inhibition of HDAC8 increases DNR sensitivity, while ectopic expression of IL-6 and IL-8 reverses siHDAC8-induced DNR sensitivity. A mechanism study demonstrated that HDAC8 could directly bind to the RELA promoter and enhance its transcription, which increases the expression of IL-6 and IL-8 in DNR-resistant AML cells [111]. HDAC8 inhibition has also been shown to significantly enhance the sensitivity of AML stem cells to cytarabine, another frontline chemotherapeutic agent used for AML [68].

#### 5.4.2. Targeted Therapy

HDAC8 mediates BRAF inhibitor (BRAFi) resistance in melanoma cells. HDAC8 is consistently overexpressed in BRAFi-resistant melanoma cell lines, and ectopic expression of HDAC8 in melanoma cells increases BRAFi tolerance. Moreover, HDAC8 overexpression significantly reduces BRAFi-induced apoptosis by suppressing the pro-apoptotic protein, Bcl-2-like protein 11 (BIM), and stabilizing anti-apoptotic protein, myeloid cell leukemia 1 (MCL1). Mass spectrometry-based phosphoproteomics revealed that HDAC8 drives BRAFi resistance through c-Jun deacetylation. HDAC8 deacetylates c-Jun and increases its transcriptional activity, subsequently enhancing receptor tyrosine kinase (RTK) and mitogen-activated protein kinase (MAPK) signaling. Introducing Lys273 to Arg (K273R) c-Jun mutant, which mimics consistent deacetylation, reduces BRAFi sensitivity in melanoma cells. Consistent with the in vitro data, co-targeting BRAF and HDAC8 showed synergistic anti-tumor effects in melanoma mouse models [81].

In melanoma and colon cancer cells, HDAC8 induces resistance to MEK1/2 inhibition by activating AKT signaling and promoting cell proliferation in the absence of MEK/ERK activation [112,113]. Indeed, Ha et al. demonstrated that MEK 1/2 inhibition by either lethal toxin (LT) or MEK1/2 inhibitor, U0126, enhances phosphorylation and activation of AKT, while PCI-34051 or siHDAC8 reverses these effects. Although the precise mechanism is yet to be clarified, it was suggested that HDAC8 activates the AKT pathway in MEK1/2 inhibition-resistant cells by increasing the expression of phospholipase C-β1 (PLCB1) and decreasing the expression of squamous cell carcinoma-1 (DESC1). PLCB1 knockdown or DESC1 overexpression mimics HDAC8 inhibition and suppresses AKT activation, preventing cell proliferation in resistant cells [113].

Mutation of internal tandem duplication (ITD) at FMS-like receptor tyrosine kinase-3 (FLT3) is one of the major mutations in AML. While several FLT3 tyrosine kinase inhibitors (TKI) have shown clinical significance, most patients experience only partial and transient responses [114]. FLT3 inhibition induces FOXO1- and FOXO3-associated HDAC8 upregulation, which inactivates p53 and drives TKI resistance in FLT3-ITD+ AML cells. Accordingly, the combination of HDAC8i, 22d, and FLT3 TKI enhances the elimination of FLT3-ITD+ AML and reduces leukemia-initiating capacity in vivo [28].

## 6. HDAC8 in Non-Cancer Disease

Pathological functions of HDAC8 go beyond cancer. HDAC8 participates in the pathogenesis of diseases such as CdLS, infectious disease, cardiovascular disease, pulmonary disease, gastrointestinal disease, hepatic disease, renal disease, neuronal disease, osteopathy, and myopathy [30,34,39,83,85,115,116,117,118,119]. Although the precise molecular mechanism is limited, HDAC8 inhibition suppresses the advancement of the diseases, demonstrating its potential as a therapeutic target.

### 6.1. CdLS and Infectious Diseases

CdLS is a severe genetic disorder characterized by distinctive facial dysmorphism, growth retardation, and cognitive impairment. Loss-of-function HDAC8 mutation is one of the five causative mutations of CdLS, and the increase in acetylated SMC3 drives both clinical and cellular features of CdLS [61,83]. HDAC8 is also associated with viral infections by regulating viral entry, replication, and host immune response [40,55,115]. Furthermore, targeting parasite-specific HDAC8 has therapeutic potential in treating parasitic diseases such as *Schistosoma mansoni* and *Mesocestoides corti* [120,121].

### 6.2. Cardiovascular Disease

Concerning cardiovascular diseases, HDAC8 is associated with the development of cardiac hypertrophy, fibrosis, inflammation, and hypertension. Ectopic expression of HDAC8 induces hypertrophy in cardiomyocytes and upregulates cardiac hypertrophy markers such as ACTA, ANP, BNP, and β-MHC. Overexpression of HDAC8 also increases the levels of phospho-AKT and phospho-GSK-3β, activating the AKT/GSK-3β pathway, which is important for growth control in the cardiovascular system [39]. In a separate study, Zhao et al. elucidated that HDAC8 stimulates cardiac hypertrophy by upregulating p38 MAPK. MAPK pathway is one of many pathways that are involved in promoting cardiac hypertrophy. Accordingly, HDAC8 inhibition by PCI-34051 suppresses p38 activation, attenuates markers of cardiac hypertrophy and fibrosis, and suppresses cardiac hypertrophy in vitro and in the isoproterenol-induced cardiac hypertrophy mouse model [122]. In the following research, HDAC8 was demonstrated to be involved in the proinflammatory RELA pathway and fibrosis-related TGF-β1-SMAD2/3 pathway via the HDAC8-ACE1 axis. Consequently, PCI-34051 treatment alleviates heart failure-related features such as cardiac hypertrophy, pulmonary congestion, fibrosis, and inflammation in vitro and in the TAC-induced heart failure model [123]. Additionally, Kee et al. demonstrated the therapeutic benefits of HDAC8 inhibition in alleviating hypertension. PCI-34051 treatment decreases high blood pressure by reducing aortic wall thickness, increasing vascular relaxation, and attenuating inflammation in the Ang II-induced hypertension mouse model [124].

### 6.3. Pulmonary Disease

HDAC8 takes part in pulmonary hypertension, fibrosis, and inflammation [86,116,125]. HDAC8 is upregulated in various tissues and cells of patients with pulmonary arterial hypertension (PAH). Additionally, HDAC8 downregulation significantly suppresses KLF2 transcription, which plays an important role in driving PAH pathogenesis [116]. HDAC8 also contributes to the pathogenesis of pulmonary fibrosis. HDAC8 expression is increased in idiopathic pulmonary fibrosis lung tissues and normal human lung fibroblasts treated with the fibrogenic cytokine, TGF-β1. siHDAC8 represses TGF-β1-induced expression of profibrotic factors such as fibronectin and type 1 collagen and increases expression of antifibrotic factors such as peroxisome proliferator-activated receptor-γ (PPARγ). Particularly, HDAC8 inhibition increases PPARγ expression by restoring TGF-β1-induced loss of H3K27ac at the *PPARG* enhancer. Moreover, NCC170, an HDAC8i, significantly decreases fibrosis score and downregulates the expression of profibrotic proteins in a bleomycin-induced pulmonary fibrosis mouse model [125].

### 6.4. Hepatic Disease

Cholestasis is a liver disease that arises when bile flow is blocked and leads to hepatotoxicity, inflammation, and fibrosis. Hepatic expression of HDAC8 is elevated in human and murine cholestasis models. SPA3014, an HDAC8i, ameliorates cholestatic liver injury by reducing necrotic damage, inflammation, and fibrosis in the liver of bile duct-ligated mice. In vitro studies demonstrated that SPA3014 attenuates hepatic fibrosis by suppressing the TGF-β signaling pathway [126]. Alcoholic hepatitis (AH) is a severe liver disease that arises from excessive alcohol consumption. In AH mice, ectopic expression of HDAC8 significantly exacerbates liver inflammation and increases proinflammatory mediators, such as TNF-α and IL-1β [34].

### 6.5. Myopathy

HDAC8 mediates various myopathies, such as Duchenne muscular dystrophy (DMD), rheumatoid arthritis (RA), and osteoarthritis (OA). DMD is a genetic disorder characterized by progressive degeneration and weakening of the muscles. HDAC8 is overexpressed in myoblasts and myotubes derived from DMD patients. PCI-34051 treatment abrogates DMD pathological phenotype by increasing myosin and reducing pro-inflammatory cytokine. Moreover, PCI-34051 treatment upregulates the acetylation of cytoskeleton proteins, especially α-tubulin, rescuing the cytoskeleton architecture of DMD myotubes [85]. RA is an autoimmune disease that causes joint destruction. Inhibition of HDAC8 with butyrate reduces the severity of arthritis by regulating T_H_17/T_reg_ balance, which is critical for maintaining immune homeostasis. Butyrate enhances the acetylation of ERRα, which suppresses its transcriptional activity, consequently upregulating the expression of carnitine palmitoyltransferase I (CPTI) and downregulating the expression of nuclear receptor subfamily 1, group D, member 1 (NR1D1). Butyrate-induced changes in CPTI and NR1D1 expression increase the number of anti-inflammatory T_reg_ cells and decrease the number of pro-inflammatory T_H_17 cells, thereby attenuating rheumatoid inflammation [127]. OA is a degenerative joint disease characterized by cartilage destruction and impaired joint function. HDAC8 is upregulated in the OA model of rat articular chondrocytes (rACs) treated with IL-1β. HDAC8 inhibition suppresses the expression of a disintegrin and metalloproteinase with thrombospondin motifs-4 (ADAMTS-4), ADAMTS-5, collagen type X (col X), and cyclooxygenase-2 (COX2), all of which are implicated in the pathogenesis of OA [128]. Furthermore, the knockdown of HDAC8 significantly increases the expression of the chondrogenic marker, SOX9, and greatly decreases the expression of degenerating marker, RUNX2, suggesting that HDAC8 can be targeted to treat osteoarthritis [42].

### 6.6. Other Diseases

HDAC8 inhibition is effective against neuroinflammation-mediated diseases. In an intrastriatal LPS-induced neuroinflammation mouse model, an HDAC8i 22d alleviates neurological deficits and reduces neuroinflammatory responses. The neuroprotection is mediated by suppressing glial activation, which is critical in the pathogenesis of neuroinflammation. Specifically, 22d decreases inflammatory enzymes and TNF-α production of microglial cells, partly resulting from the regulation of STAT1/3 and AKT activity [119]. Fibrous dysplasia (FD) is a genetic disorder characterized by the development of fibrous tissue in place of normal bone. *GNAS* mutations are frequently observed in FD cases and are considered to be the leading cause of the disease. HDAC8 is overexpressed in FD BMSCs due to *GNAS* mutation that activates the cAMP-CREB-HDAC8 axis. Upregulation of HDAC8 is involved in FD pathogenesis by enhancing proliferation and decreasing osteogenesis in BMSCs, which is mediated by HDAC8-induced repression of p53 and RUNX2, respectively. Accordingly, pharmacological inhibition and genetic downregulation of HDAC8 enhance the osteogenic potential in vitro and in vivo [30].

## 7. HDAC8-Selective Inhibitors

### 7.1. PCI-34051

PCI-34051 is an indole-based HDAC8i (IC_50_ = 56 nM) [129] that was developed by Pharmacyclics Inc. in 2008 (Table 3). It contains a metal-chelating hydroxamic acid group that potentially binds to metalloenzymes. It has more than 200-fold specificity for HDAC8 inhibition over other class I HDACs as well as HDAC6 and HDAC10 [49].

The combination effects of PCI-34051 with other therapeutic agents have been studied for anti-cancer effects. PCI-34051 exerts synergistic effects when combined with chemotherapy, such as paclitaxel in TNBC [57] and cytarabine in AML [130]. PCI-34051 also shows synergism with targeted therapy. PCI-34051 enhances anti-cancer effects when treated with ALK/MET inhibitor, crizotinib, in neuroblastoma cells, [131] nicotinamide phosphoribosyltransferase inhibitor, KPT-9274, in AML cells, [132] and HDAC6 inhibitor, ACY-241, in ovarian cancer cells [66]. Lastly, PCI-34051, in combination with tumor necrosis factor-related apoptosis-inducing ligand (TRAIL) variants, has synergistic apoptotic effects in TRAIL-resistant colon cancer cells (DLD-1, WiDr) [133].

### 7.2. Aryl Hydroxamic Acids

Based on the crystal structure of HDAC8, six aryl hydroxamic acids (compounds **1**–**6**) were synthesized in 2007. Unlike other HDAC inhibitors that have the canonical zinc binding group-linker-cap group, aryl hydroxamic acids are linkerless and target the unique sub-pocket located adjacent to M274 of the active site [134].

### 7.3. NBM-BMX (BMX)

BMX, synthesized from osthole [135], is the first and only HDAC8i that has entered the clinical trial. Currently, it is in a clinical trial for advanced cancer through oral administration (NCT03808870, NCT03726294). BMX has been reported to improve learning and memory in rat models [135] and to help restore TMZ sensitivity in glioblastoma [109].

### 7.4. WK2-16

Several ortho-aryl N-hydroxycinnamides were synthesized to target HDAC8 specifically. Among those compounds, (E)-N-hydroxy-4-methoxy-2-(biphenyl-4-yl) cinnamide (WK2-16 or 22d) effectively inhibits HDAC8 with an IC_50_ of 27.2 nM. WK2-16 exhibits greater anti-proliferative effects than PCI-34051 in human lung cancer cells (CL1-5, H1299, A549) without cytotoxicity in normal IMR-90 cells. In CL1-5 cells, the effect of WK2-16 is similar to that of SAHA [129]. Moreover, WK2-16 effectively synergizes with quizartinib (AC220), a second-generation TKI [136], in FLT3-ITD+ AML to exhibit anti-leukemia effects in vitro and in vivo. [28].

### 7.5. Dual Inhibitors

Dual inhibition of HDAC8 and MMP-2 is a potent anti-cancer therapeutic strategy for solid and hematological cancers [137,138]. The first chemical that inhibited both HDAC8 and MMP-2 was derived from an L-tyrosine scaffold. Hydroxamate compounds of L-tyrosine derivatives selectively inhibit MMP-2 and HDAC8 with equal potency as NNGH and SAHA, which are well-known inhibitors of MMP and HDAC, respectively [139]. In another study, a pharmacophore model was implemented to design an HDAC8/MMP-2 dual inhibitor. The newly synthesized molecule inhibits HDAC8 activity by interacting with specific amino acid residues (Tyr306, Ile34, Phe152, His143, and Cys153) at the acetate release channel. The inhibitor significantly suppresses cell migration and invasion of the A549 human lung cancer cell line [137]. 

**Table 3 cells-11-03161-t003:** Chemical structures of HDAC8-selective inhibitors.

PCI-34051	Aryl Hydroxamic Acids Compound 6
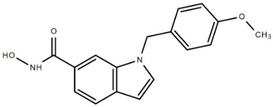	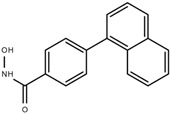
**BMX**	**WK2-16**
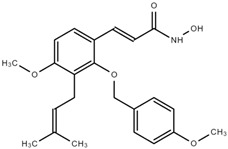	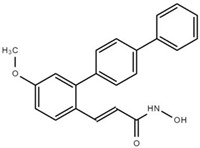

## 8. Conclusions

Since its discovery in 2000, intensive studies have been conducted to characterize the structure and function of HDAC8 [11,12,13]. Functional studies of HDAC8 revealed that it plays a pivotal role in the pathogenesis of diseases such as cancer, CdLS, and infectious diseases and raised the need to find selective inhibitors that specifically target HDAC8. A study in 2004 identified the crystal structure of HDAC8 and accelerated the development of HDAC8is [25]. Since then, various HDAC8is have been developed, and a comprehensive review of HDAC8is has recently been reported [140]. Despite the substantial number of inhibitors developed, BMX is the only HDAC8i that has entered the clinical trial. Thus, further investigation to find an effective HDAC8i is needed. Recently, HDAC8-targeting proteolysis targeting chimeras (PROTACs) have been designed and have shown significant selectivity towards HDAC8, suggesting a novel therapeutic strategy [141,142].

Studies of HDAC8 are heavily concentrated on its role in cancer progression. HDAC8 is upregulated in cancer and is associated with tumor cell proliferation, apoptosis, metastasis, immune evasion, and drug resistance. Accordingly, HDAC8is have been tested alone or in combination in various types of cancer and have shown anti-cancer effects in vitro and in vivo. Recent studies have expanded our knowledge of HDAC8 in non-cancer diseases such as cardiovascular disease, pulmonary disease, and myopathy. HDAC8 was overexpressed in these diseases and associated with the upregulation of disease-related biomarkers. However, the molecular mechanisms underlying the pathogenesis of these diseases are poorly understood. Therefore, further studies are required to increase our understanding of HDAC8 in non-cancer diseases. 

Here, we focused on the pathological function of HDAC8. However, some important discoveries have been made in the physiological functions of HDAC8. For instance, HDAC8 supports smooth muscle contraction by interacting with substrates such as α-SMA, cortactin, and cofilin [20,21,77]. HDAC8 is also involved in skeletal muscle and neural differentiation [143,144,145] as well as stem cell maintenance [50,146]. Nevertheless, the limited number of studies on the physiological function of HDAC8 suggests more studies should be conducted to explore novel substrates and functions of HDAC8 in normal physiology. Multiple mechanisms regulate the expression and activity of HDAC8, and in turn, HDAC8 is involved in important cellular processes such as cell cycle, migration, and differentiation. The latest research on HDAC8 highlights the therapeutic potential of targeting HDAC8 to overcome diseases. Thus, further investigations on HDAC8 substrates, as well as effective HDAC8is, are required.

## Figures and Tables

**Figure 1 cells-11-03161-f001:**
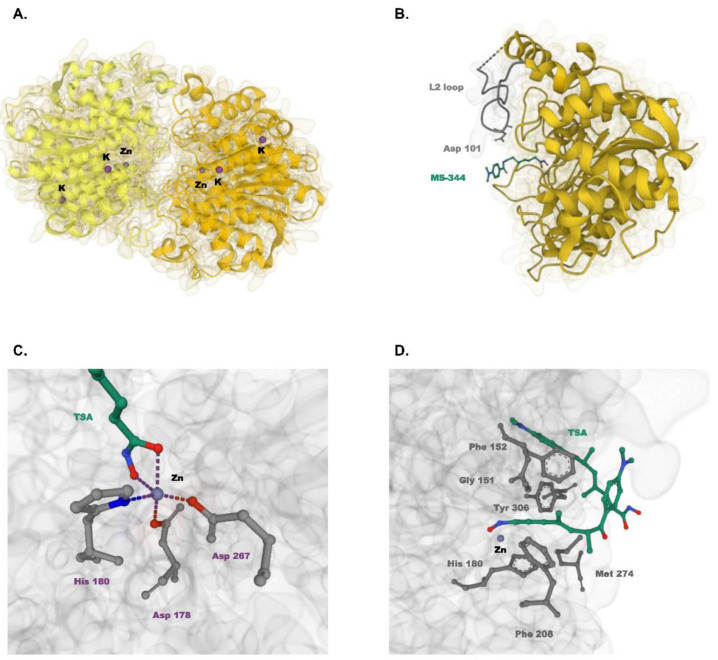
Crystal structure of HDAC8. (**A**) Head-to-head dimeric arrangement of HDAC8 (PDB ID: 2V5W [27]). Two monomers of HDAC8 are shown in light and dark yellow. Zinc and potassium are represented as grey and purple spheres, respectively (**B**) Liganded form of HDAC8 (PDB ID: 1T67 [15]). L2 loop and MS-344 are colored grey and green, respectively. (**C**) Catalytic machinery of HDAC8 (PDB ID: 1T64 [15]). Zinc ion is coordinated to three HDAC8 residues and two oxygen atoms of TSA. The zinc coordination shell is indicated by dashed lines. (**D**) Active site of HDAC8 (PDB ID: 1T64 [15]). Six key residues form the hydrophobic tunnel that occupies two TSA molecules.

**Figure 2 cells-11-03161-f002:**
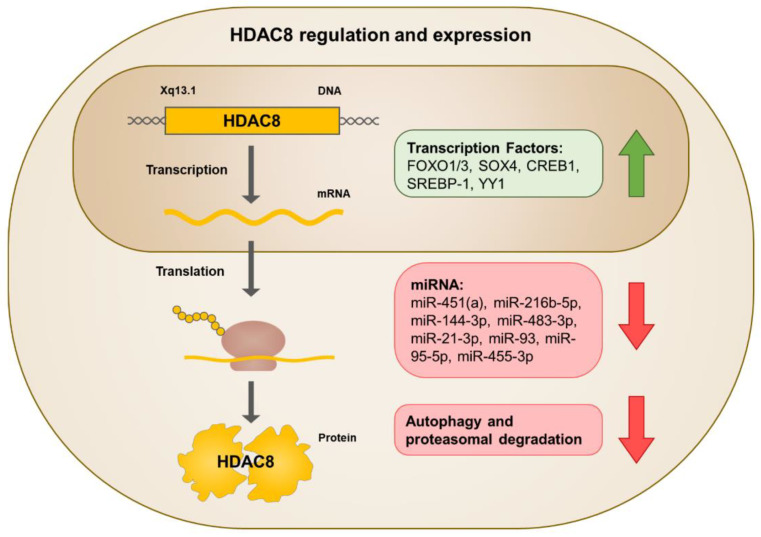
Schematic representation of human HDAC8 and its regulation. HDAC8 is upregulated by transcription factors and downregulated by miRNA, autophagy, and proteasomal degradation.

**Figure 3 cells-11-03161-f003:**
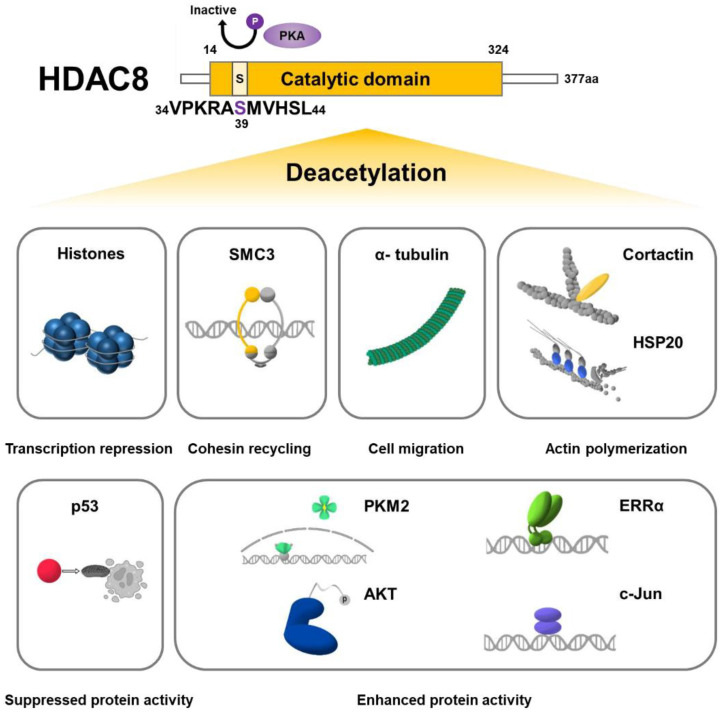
Schematic representation of histone and non-histone substrates of HDAC8. HDAC8 deacetylates histones and suppresses transcription of target genes. HDAC8 also deacetylates non-histone substrates such as SMC3, α-tubulin, cortactin, HSP20, p53, PKM2, AKT, ERRα, and c-Jun.

**Figure 4 cells-11-03161-f004:**
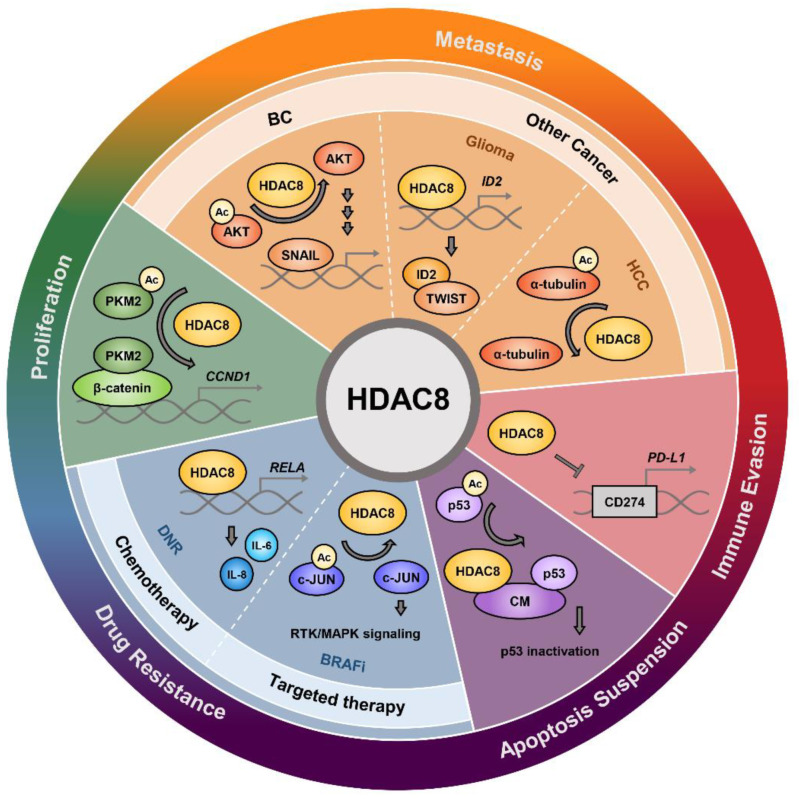
Schematic representation of pathological functions of HDAC8 in cancer. HDAC8 promotes tumor growth by enhancing tumor cell proliferation and suppressing apoptosis. HDAC8 also enhances metastasis and is involved in drug resistance and immune evasion.

**Table 1 cells-11-03161-t001:** HDAC8-mediated transcription repression of target genes.

Gene	Region	Histone Substrate	Cells	Reference
IFNB1	promoter	H3/H4	Macrophages	[55]
SOCS1/3	promoter	H3/H4	Erythroleukemia cells	[56]
SIRT7	promoter	H4	BC cells	[57]
MAP2K3	promoter	H3K9 and H3K27	Keratinocytes	[58]
CCL4	enhancer	H3K27	HCC cells	[54]
ID2	enhancer	H3K27	HCC cells	[59]
BNIP3	N/A	H3K27	TIR cells	[60]
MLN64	N/A	H3K27	TIR cells	[60]

## Data Availability

Not applicable.

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
