# Peer review of "Pathological Role of HDAC8: Cancer and Beyond"

_cells, 2022, doi:10.3390/cells11193161_

Round 1
Reviewer 1 Report
The article titled “Pathological role of HDAC8: Cancer and Beyond, by Kim et al., the authors do an extensive review on HDAC8 de-acetylase in human cells. This includes a description of the enzyme, which targets have been described so far, and how modifications of HDAC8 affect its function. They also briefly review how the expression of the corresponding gene is regulated; analyze the de-acetylation of K9 and K27 of histone 3 (H3) as well as targets in the cytoplasm.
They then the authors discuss the role that HDAC8 may have in the generation and maintenance of the cancerous phenotype, in particular when HDAC8 is over-expressed and the relationships that exist with it and metastasis and drug resistance. Therefore, they explain why HDAC8 is an interesting target for treating some types of cancer and they describe very well the inhibitors that have been developed to inhibit HDAC8 to date.
In general, it is a good review on HADC8 that gives a broad overview of the role of this enzyme in the cell and in the generation of cancer. However, there is a problem about reporting a new review on the HDAC8. This is that recently there have been reviews on HDAC8 that cover the different aspects that are treated by Kim, et al. Therefore, I do not think a new review on HDAC8 and cancer is appropriate at this time.
Here are some examples of recent reviews on the HDCA8:
A Therapeutic Perspective of HDAC8 in Different Diseases: An Overview of Selective Inhibitors.
Fontana A, Cursaro I, Carullo G, Gemma S, Butini S, Campiani G. Int J Mol Sci. 2022 Sep 2;23(17):10014. doi: 10.3390/ijms231710014. PMID: 36077415 Review.
Phenotypes of Cornelia de Lange syndrome caused by non-cohesion genes: Novel variants and literature review.
Shangguan H, Chen R. Front Pediatr. 2022 Jul 22;10:940294. doi: 10.3389/fped.2022.940294. eCollection 2022.
The clinical significance of histone deacetylase-8 in human breast cancer.
Rahmani G, Sameri S, Abbasi N, Abdi M, Najafi R. Pathol Res Pract. 2021 Apr;220:153396. doi: 10.1016/j.prp.2021.153396. Epub 2021 Mar 1. PMID: 33691240 Review.
Structure-Based Inhibitor Discovery of Class I Histone Deacetylases (HDACs).
Luo Y, Li H. Int J Mol Sci. 2020 Nov 22;21(22):8828. doi: 10.3390/ijms21228828. PMID: 33266366 Free PMC article. Review
Suggestions for authors:
1.- In the introduction, the authors mention that histone modifications, including acetylation, are part of epigenetic inheritance. So far there is no publication showing that histone acetylation is involved in epigenetic inheritance. In fact, it has recently been shown that before mitosis all histone acetylation marks are erased, since this is necessary for the mitotic chromosomes to condense. I suggest not using the term epigenetic marks on histones.
2.-It would greatly enrich the review to mention how conserved HDCA8 is in evolution. Is it specific to vertebrates? or is it present in other organisms?
3.-Also, mention in which tissues HDCA8 is expressed in humans and what happens during development in different experimental models. Is there a KO in mice? It is an essential gene?
4.-The HDCA's in order to de-acetylate histones in specific regions of the genome require that they be driven by other factors. It would be important to mention that it is known about those who drive HDCA8 to its targets in the genome. For example, is it part of some protein complex that represses transcription?
Author Response
Reviewer 1
The article titled “Pathological role of HDAC8: Cancer and Beyond, by Kim et al., the authors do an extensive review on HDAC8 de-acetylase in human cells. This includes a description of the enzyme, which targets have been described so far, and how modifications of HDAC8 affect its function. They also briefly review how the expression of the corresponding gene is regulated; analyze the de-acetylation of K9 and K27 of histone 3 (H3) as well as targets in the cytoplasm.
They then the authors discuss the role that HDAC8 may have in the generation and maintenance of the cancerous phenotype, in particular when HDAC8 is over-expressed and the relationships that exist with it and metastasis and drug resistance. Therefore, they explain why HDAC8 is an interesting target for treating some types of cancer and they describe very well the inhibitors that have been developed to inhibit HDAC8 to date.
In general, it is a good review on HADC8 that gives a broad overview of the role of this enzyme in the cell and in the generation of cancer. However, there is a problem about reporting a new review on the HDAC8. This is that recently there have been reviews on HDAC8 that cover the different aspects that are treated by Kim, et al. Therefore, I do not think a new review on HDAC8 and cancer is appropriate at this time.
Here are some examples of recent reviews on the HDCA8:
A Therapeutic Perspective of HDAC8 in Different Diseases: An Overview of Selective Inhibitors. Fontana A, Cursaro I, Carullo G, Gemma S, Butini S, Campiani G. Int J Mol Sci. 2022 Sep 2;23(17):10014. doi: 10.3390/ijms231710014. PMID: 36077415 Review.
Phenotypes of Cornelia de Lange syndrome caused by non-cohesion genes: Novel variants and literature review. Shangguan H, Chen R. Front Pediatr. 2022 Jul 22;10:940294. doi: 10.3389/fped.2022.940294. eCollection 2022.
The clinical significance of histone deacetylase-8 in human breast cancer. Rahmani G, Sameri S, Abbasi N, Abdi M, Najafi R. Pathol Res Pract. 2021 Apr;220:153396. doi: 10.1016/j.prp.2021.153396. Epub 2021 Mar 1. PMID: 33691240 Review.
Structure-Based Inhibitor Discovery of Class I Histone Deacetylases (HDACs).Luo Y, Li H. Int J Mol Sci. 2020 Nov 22;21(22):8828. doi: 10.3390/ijms21228828. PMID: 33266366 Free PMC article. Review
Suggestions for authors:
Response: We thank the reviewer for taking the time and effort to revise our manuscript entitled “Pathological role of HDAC8: Cancer and Beyond.” In regards to the general comment, we acknowledge that multiple review papers have been published on HDAC8 in recent years. However, most of these reviews are heavily concentrated on either specific disease such as Cornelia de Lange syndrome (CdLS) or breast cancer or focused on selective inhibitors. Our review provides a comprehensive overview of the pathological function of HDAC8 in cancer progression. Therefore, we believe that this manuscript will allow the readers to gain a better understanding of HDAC8.
1.- In the introduction, the authors mention that histone modifications, including acetylation, are part of epigenetic inheritance. So far there is no publication showing that histone acetylation is involved in epigenetic inheritance. In fact, it has recently been shown that before mitosis all histone acetylation marks are erased, since this is necessary for the mitotic chromosomes to condense. I suggest not using the term epigenetic marks on histones.
Response: We are thankful for your insightful comment. When we explained that “accumulation of epigenetic alterations is heritable and drives the malignant transformation of cells,” we were not intending to specify histone acetylation but rather refer to the long lasting concept that epigenetic modifications such as DNA methylation and histone modifications can be passed to the next generation. {Skvortsova, Ksenia, Nicola Iovino, and Ozren Bogdanović. "Functions and mechanisms of epigenetic inheritance in animals." Nature reviews Molecular cell biology 19.12 (2018)}. Since histone acetylation is a critical epigenetic modification, we felt it was necessary to describe this concept in the introduction part of our manuscript before we went on to elaborate on HDAC8.
2.-It would greatly enrich the review to mention how conserved HDCA8 is in evolution. Is it specific to vertebrates? or is it present in other organisms?
Response: We thank the reviewer for their valuable comments. HDACs are members of an enzyme family and are found in plants, animals, fungi, archaebacteria and eubacteria {Leipe, Detlef D et al.” Histone deacetylases, acetoin utilization proteins and acetylpolyamine amidohydrolases are members of an ancient protein superfamily." Nucleic acids research 25.18 (1997)}. In fact, classification of HDACs have been made based on the sequence similarity to yeast factors and cofactor dependency (Line 45-47). However, the study of HDAC8 expression in other organisms is limited. Recent studies have identified parasite specific HDAC8 in Schistosoma Mansoni and Mesocestoides Corti (Line 424-425). We agree with the reviewer that it would be very insightful to mention conserved HDAC8 in other species, yet information regarding such topic was very limited.
3.-Also, mention in which tissues HDCA8 is expressed in humans and what happens during development in different experimental models. Is there a KO in mice? It is an essential gene?
Response: Ubiquitous expression of HDAC8 has been observed in human tissues {Waltregny, David, et al. "Expression of histone deacetylase 8, a class I histone deacetylase, is restricted to cells showing smooth muscle differentiation in normal human tissues." The American journal of pathology 165.2 (2004)}. HDAC8 is particularly involved in smooth muscle contraction as well as the skeletal muscle and neural differentiation (Line 584-586). In this manuscript we have elaborated the role of HDAC8 in cancer development as well as summarize the types of cancer that demonstrated high expression of HDAC8 (Line 226-229). Furthermore, we mentioned multiple experimental models including mouse models where HDAC8 expression played a critical role in cancer development and described that genetic ablation or pharmacological inhibition showed anti-cancer effects. Several studies have designed HDAC8 knockout cells {Katayama, Syouichi et al. “HDAC8 regulates neural differentiation through embryoid body formation in P19 cells.” Biochemical and biophysical research communications vol. 498,1 (2018); Zhang, Pu et al. “Targeting DNA Damage Repair Functions of Two Histone Deacetylases, HDAC8 and SIRT6, Sensitizes Acute Myeloid Leukemia to NAMPT Inhibition.” Clinical cancer research: an official journal of the American Association for Cancer Research vol. 27,8 (2021)}. However, global deletion of HDAC8 leads to perinatal lethality caused by skull instability and thus is less common in experimental settings {Haberland, Michael, et al. "Epigenetic control of skull morphogenesis by histone deacetylase 8." Genes & development 23.14 (2009)}. Nevertheless, Singh et al. designed a conditional HDAC8 KO mice model to demonstrate the role of HDAC8 in female fertility {Singh, Vijay Pratap et al. “Oocyte-specific deletion of Hdac8 in mice reveals stage-specific effects on fertility.” Reproduction (Cambridge, England) vol. 157,3 (2019)}. However, such paper was not mentioned in our manuscript as it was beyond the scope of this review.
4.-The HDCA's in order to de-acetylate histones in specific regions of the genome require that they be driven by other factors. It would be important to mention that it is known about those who drive HDCA8 to its targets in the genome. For example, is it part of some protein complex that represses transcription?
Response: Several mechanisms that explain how HDACs repress transcription have been suggested over the years. Primarily, deacetylating the lysine residues increases the positive charge of core histones and strengthens histone tail-DNA interaction which blocks access of transcriptional machinery to DNA (Line 35-37). Furthermore, deacetylated lysine residues can recruit other epigenetic modulators such as histone methyltransferases. Methylated lysine residues can then be recognized by HP1 which promotes heterochromatin formation, further suppressing gene transcription. These features are not specific to HDAC8, thus the mechanism was not described in detail in this manuscript but briefly mentioned in the introduction part.
Reviewer 2 Report
Authors J.-Y. Kim, et al. present a literature review of the HDAC8 enzyme with a focus on its function under normal physiological conditions as well as its pathological roles in cancer as well as other disease states and, consequently, its potential as a therapeutic target. The manuscript is well organized and the writing is very good. The handling of the topic is current with many of the cited studies coming from the previous five years. With the exception of the relative lack of illustrative figures (noted below), the review is very good.
Major criticism
The manuscripts suffers for its meager use of figures. Furthermore, the figures that are included do not contribute significantly to the message of the manuscript. Much of Figure 1, for example, is simply a symbolic representation of the Central Dogma of Molecular Biology, and Figure 3 does nothing except list five of the Hallmarks of Cancer (and, consequently, the subheadings of that section of the paper) in which a role for HDAC8 has been identified. There are several other points throughout the manuscript where a well conceived figure could illustrate and/or clarify what is being written. For example, the second paragraph of page 2 describes the catalytic mechanism of HDAC8 within the context of the x-ray crystal structure of human HDAC8. This would naturally lend itself to a figure--either something derived from the atomic coordinates of the x-ray crystal structure or a schematic diagram that illustrates the points addressed in the description of the enzyme's catalytic mechanism. Additionally, in the introduction to the discussion of substrates of HDAC8, it is mentioned that, "...the development of biochemical investigation techniques potentates the discovery of other HDAC8 substrates. (lines 114-116). Would it make sense to develop this and discuss (perhaps with a figure) what approaches have been used and what new approaches might be applied to conclusively identify HDAC8 substrate proteins?
Minor criticisms
Page 2, line 72: The first sentence of this paragraph on HDAC8 structure needs to be rewritten. Perhaps something like, "HDAC8 is the first human HDAC to be crystallized and have its three-dimensional structure experimentally determined by x-ray crystallography."
Page 14, line 549: The first sentence of this paragraph needs editing. Perhaps, "Dual inhibition of HDAC8 and MMP-2 is a potent anti-cancer therapeutic strategy for solid and hematological cancers."
Page 15: The molecule structures depicted in Table 3 are very small and difficult to see. Also, this might be a good place to return to the HDAC8 structure and mechanism to illustrate (with a figure or schematic) how different classes of inhibitors function to block HDAC8 activity and what this might suggest for future drug development strategies.
Author Response
Reviewer 2
Authors J.-Y. Kim, et al. present a literature review of the HDAC8 enzyme with a focus on its function under normal physiological conditions as well as its pathological roles in cancer as well as other disease states and, consequently, its potential as a therapeutic target. The manuscript is well organized and the writing is very good. The handling of the topic is current with many of the cited studies coming from the previous five years. With the exception of the relative lack of illustrative figures (noted below), the review is very good.
Major criticism
The manuscripts suffers for its meager use of figures. Furthermore, the figures that are included do not contribute significantly to the message of the manuscript. Much of Figure 1, for example, is simply a symbolic representation of the Central Dogma of Molecular Biology, and Figure 3 does nothing except list five of the Hallmarks of Cancer (and, consequently, the subheadings of that section of the paper) in which a role for HDAC8 has been identified. There are several other points throughout the manuscript where a well conceived figure could illustrate and/or clarify what is being written. For example, the second paragraph of page 2 describes the catalytic mechanism of HDAC8 within the context of the x-ray crystal structure of human HDAC8. This would naturally lend itself to a figure-either something derived from the atomic coordinates of the x-ray crystal structure or a schematic diagram that illustrates the points addressed in the description of the enzyme's catalytic mechanism. Additionally, in the introduction to the discussion of substrates of HDAC8, it is mentioned that, "...the development of biochemical investigation techniques potentates the discovery of other HDAC8 substrates. (lines 114-116). Would it make sense to develop this and discuss (perhaps with a figure) what approaches have been used and what new approaches might be applied to conclusively identify HDAC8 substrate proteins?
Response: We thank the reviewer for their valuable comments. The main purpose of Figure 1 was to illustrate the different ways in which HDAC8 expression levels could be regulated. The Central Dogma was shown to visualize how various factors (shown in the left side of the figure) could affect the transcription, translation, and degradation of HDAC8. We have edited Figure 3 so that the figure would give more information than just list the five hallmarks of cancer. We have illustrated representative mechanisms where HDAC8 plays a critical role in metastasis, proliferation, apoptosis suppression, drug resistance, and immune evasion. Furthermore, as suggested by the reviewer, we have added a crystal structure of human HDAC8 to enhance the understanding of the structural description provided in the manuscript (Figure 1). As for the development of biochemical techniques, it is difficult to specify which biochemical approach attributed to the discovery of specific HDAC8 substrates. Most of the advancements made in biotechnology such as the X-ray crystallography technique, CRISPR-Cas9 genome editing technique, development of specific antibodies or inhibitors, etc., collectively contributed to the discovery of the substrates. Thus, we were unable to provide a figure that would illustrate which new approach conclusively led to the identification of substrate proteins. We are very thankful to the reviewer for the insightful remarks that greatly enhanced the quality of this manuscript.
Minor criticisms
Page 2, line 72: The first sentence of this paragraph on HDAC8 structure needs to be rewritten. Perhaps something like, "HDAC8 is the first human HDAC to be crystallized and have its three-dimensional structure experimentally determined by x-ray crystallography."
Response: Thank you for the insightful comment. We have substituted the previous sentence with the recommended one.
Page 14, line 549: The first sentence of this paragraph needs editing. Perhaps, "Dual inhibition of HDAC8 and MMP-2 is a potent anti-cancer therapeutic strategy for solid and hematological cancers."
Response: We agree with the reviewer's comment. We have changed the sentence with the recommended version.
Page 15: The molecule structures depicted in Table 3 are very small and difficult to see. Also, this might be a good place to return to the HDAC8 structure and mechanism to illustrate (with a figure or schematic) how different classes of inhibitors function to block HDAC8 activity and what this might suggest for future drug development strategies.
Response: We thank the reviewer for the comment. As suggested, we have enlarged the structure of the inhibitors in our manuscript. In this review, we aimed to focus on the pathological function of HDAC8 especially in cancer and minimize the description on inhibitors since numerous recent reviews have been dedicated in organizing the different types of HDAC8i {Amin, Sk Abdul, Nilanjan Adhikari, and Tarun Jha. "Structure-activity relationships of HDAC8 inhibitors: Non-hydroxamates as anticancer agents." Pharmacological Research 131 (2018); Adhikari, Nilanjan, Sk Abdul Amin, and Tarun Jha. "Selective and nonselective HDAC8 inhibitors: a therapeutic patent review." Pharmaceutical Patent Analyst 7.6 (2018); Banerjee, Suvankar, et al. "Histone deacetylase 8 (HDAC8) and its inhibitors with selectivity to other isoforms: An overview." European Journal of Medicinal Chemistry 164 (2019). Instead, we introduced some of the well-known and frequently utilized inhibitors that were shown to be effective in cancer.
Reviewer 3 Report
I have read with great interest this paper where the Authors report a very interesting review regarding the potential therapeutic role of HDAC8 in human cancers and other diseases. For such a reason the article should be suitable for publication after minor corrections.
1- The author are invited to added a section about methodology adopted for the data collection, extraction and organization
2- The authors should carefully proof-read the entire manuscript to minimize typographical errors, especially with spellings, punctuations, unnecessary capitalizations, superscripts/subscripts, spaces, and units, as well as to ensure uniform expression of various special characters and abbreviations, terms, and phrases.
Author Response
Reviewer 3
I have read with great interest this paper where the Authors report a very interesting review regarding the potential therapeutic role of HDAC8 in human cancers and other diseases. For such a reason the article should be suitable for publication after minor corrections.
1- The author are invited to added a section about methodology adopted for the data collection, extraction and organization
Response: We thank the reviewer for their valuable comments. This review was concentrated in giving a thorough understanding of the pathological function of HDAC8. Thus, we analyzed the discoveries made in HDAC8 that were accessible via PubMed and summarized the findings that were relevant to the topic of our manuscript. Our review did not collect, extract, and reorganize clinical data or involve big data analysis. Thus, there were no specific methods that requires further elaboration.
2- The authors should carefully proof-read the entire manuscript to minimize typographical errors, especially with spellings, punctuations, unnecessary capitalizations, superscripts/subscripts, spaces, and units, as well as to ensure uniform expression of various special characters and abbreviations, terms, and phrases.
Response: Thank you for the insightful comment. We have carefully proof-read the entire manuscript for errors.
Round 2
Reviewer 1 Report
I consider that the authors adequately answered my concerns and in this new version the manuscript can be accepted for publication.